# Factors Determining the Competitive Strategic Positions of the SMEs in Asian Developing Nations: Case Study of SMEs in the Agricultural Sector in Sri Lanka

RPIR Prasanna [1,*], JMHM Upulwehera [1], BDTN Senarath [2], GAKNJ Abeyrathne [3], PSK Rajapakshe [4], JMSB Jayasundara [4], EMS Ekanayake [5] and Sisira Kumara Naradda Gamage [1]

1 Department of Economics, Rajarata University of Sri Lanka, Mihintale 50300, Sri Lanka; harshanamiyuranga@gmail.com (J.U.); naraddagamage@ssh.rjt.ac.lk (S.K.N.G.)
2 Department of Operations Management, University of Peradeniya, Peradeniya 20400, Sri Lanka; dinusenarath95@gmail.com
3 Department of Census and Statistics, Research and Special Studies Division, Battaramulla 10120, Sri Lanka; abeyrathne.rjt@ssh.rjt.ac.lk
4 Department of Environmental Management, Rajarata University of Sri Lanka, Mihintale 50300, Sri Lanka; pskr75@ssh.rjt.ac.lk (P.R.); jmsb1610@ssh.rjt.ac.lk (J.J.)
5 Department of Social Sciences, Rajarata University of Sri Lanka, Mihintale 50300, Sri Lanka; emseka-nayake@ssh.rjt.ac.lk
* Correspondence: prasannarjt@ssh.rjt.ac.lk; Tel.: +94-71-072-3083

**Abstract:** Economic globalization has rapidly intensified the competition among businesses. Therefore, it is pivotal that SMEs follow competitive strategic positions and adopt strategic methods in order to confront the various challenges in this era. This study assessed the factors that determine the competitive strategic position of SMEs in the developing nations of Asia by considering the agro-based SMEs in Sri Lanka as a case study. By using primary data of 463 SMEs, the study estimated a binary logistic regression model to deal with the research subject. The findings revealed seven significant strategic variables: innovation in product and marketing, business exhibition in the local setting, gender, strategic market location, sustainable business practices, marketing efficiency, and business reputation and superior services. Moreover, the study identified and commented on seven insignificant variables: specialization in production, experience in the business field, experience in attending business exhibitions at overseas setting, credit market accessibility, provision of high-quality products, research and development, and strategic firm location, which are relevant in developed countries. Therefore, the government and policymakers must initiate measures to establish a more favorable business environment for SMEs to gain competitive advantage from these variables in the near future to permit an ameliorated and strong SME sector in Sri Lanka.

**Keywords:** agro-based industries; competitive advantage; competitive strategic position; economic globalization; SMEs; survival strategies

## 1. Introduction

Economic globalization involves a broad range of opportunities, procedures, and problems related to spreading economic activities among countries worldwide (Pyle 2001). Globalization is the diminution or elimination of state-enforced restrictions on exchanges across borders and is the increasingly integrated and complex global system of production and exchange which emerged as a result (Palmer 2002). The concept of globalization explains how the economic culture and political systems have been transferred in dependency terms. Today, world economic globalization is recognized as an opportunity for developed and developing nations to enhance economic prosperity by taking part in global trade (Ahmedova 2015; WTO 2016; Prasanna et al. 2019).

As Mundim et al. (2000) emphasized, world economic globalization is recogized as a remarkable source of opportunities and threats for economic activities. This has positive and negative effects on the business world. According to Mago et al. (2013), globalization has increased international trade between countries, increasing trade by growing businesses and revenue. At the same time, globalization has increased the interdependence of the economies. Nistor (2007) suggested that globalization creates positive opportunities like giving the opportunity for emerging nations to develop in a rapid growth due to their ability to access new technologies, information, direct investments and loans, products, and the high mobility of all production factors. In contrast, Mago et al. (2013) elaborated that globalization creates interdependence between countries, which can have a negative impact as depression in one economy can impact the economies of other countries.

Small and medium enterprises (SMEs) are considered the backbone of any country since they help to increase its economic growth. Dominguez and Mayrhofer (2017) indicated that SMEs could face both challenges and opportunities as a result of global economy changes in the era of economic globalization. SMEs are challenged to remain competitive in large markets from the effects of globalization (Maarof and Mahmud 2016). Therefore, compared to other types of organizations, SMEs receive more attention since they make a significant contribution to the economies of both developing and developed countries (Şener et al. 2014; Asare et al. 2015; Bilal and Al Mqbali 2015; Auzzir et al. 2018).

Many researchers have focused on the challenges the SMEs face due to globalization as it generates obstacles when they generate business on the global level. Ren et al. (2015) stated that the factors that distinguish SMEs from large enterprises, such as having limited resources, entry barriers at the global level, and having less innovative capabilities, the internationalization strategies of SMEs are different. Moreover, Şener et al. (2014) showed that although large-scale enterprises can overcome the challenges arising from globalization, they cannot overcome them due to the specific features of SMEs. On the other hand, Singh et al. (2009) argued that globalization would generate the opportunity for SMEs to integrate with large-scale enterprises. Therefore, SMEs have both positive and negative impacts on economic globalization and, at the same time, globalization creates both challenges and opportunities for SMEs.

Competitive positioning refers to how an organization or a firm differentiates itself from the competition and acquires market value. As a result, these organizations should inevitably determine their competitive strategic positions. Knowing the competitive strategic positioning allows organizations to make tactical decisions about whether they need to keep or strengthen their following positions, or whether they need to withdraw from the market. Consequently, the understanding of the competitive positions of an organization and its competitors is crucial since competitors are described as organizations that can interrupt a firm's market goals and significant moderators of a firm's performance (Day 1984). Thus, they are recognized as the most important factors in competitive strategies (Porter 1985). Therefore, SMEs must use tools such as Porter's generic strategies, Profitability Impact of Marketing Strategies (PIMS) model, the BCG (Boston Consulting Group) matrix, and the Opportunities, Strengths, Weaknesses and Threat analysis (SWOT) to identify the factors determining their competitive strategic positions.

During the last few decades, competition in the business world intensified with the acceptance of free-market principles or acknowledgment of principles of economic liberalization by many countries. In 1995, the World Trade Organization (WTO) was formed, extending the standards of the GATT further facilitating free trade. Neo-classical economic principles encourage the movement of economic resources across borders, specifically the capital and skilled labor with managerial abilities. It is generally accepted that economic competition directs the resource allocation from less productive areas to productive areas of the economies enabling economies to be more efficient. However, a growing number of studies cite vulnerability impacts of intensified competition in the economic globalization era on different sectors of the economies of the developing countries and thereby the issue of sustainable growth. Small and Medium Enterprises (SMEs) are among the

sectors which became vulnerable to economic competition. The literature cites the closing down of many SMEs within a shorter period of their business commencement due to the competitive challenges.

By releasing the World Trade Report by the WTO in 2016 on the theme "Leveling the Trading Field for SMEs" the WTO emphasized the need for change in an international setting to improve SMEs' participation in global trade. Recently, scholars in business analysis have identified three competitive challenges—global challenges, technology challenges, and sustainability challenges—facing the business firms (Noe et al. 2017). Various studies provide further evidence over these three competitive challenges specifically facing the SMEs (Prasanna et al. 2019; Naradda Gamage et al. 2020; Jayasundara et al. 2019) but only a limited number of studies have investigated how SMEs improve their survival chance in facing these competitive challenges in the globalized economy. Thus, the central aim of this study is to examine the factors determining the competitive strategic position of the SMEs in the developing nations of Asia by considering the agro-based SMEs in Sri Lanka as a case study.

### 1.1. Roles and Specific Features of the SMEs

The SME sector is generally accepted as the backbone of an economy because it contributes economies to be more resilient and stable since SMEs handle a significant proportion of business activities. For instance, Yoshino and Taghizadeh-Hesary (2016) reported that SMEs account for 90% of businesses in Asia, and Bilal and Al Mqbali (2015) reported 95% of SMEs share in the business worldwide. The literature primarily demonstrated the sector's contribution in empowering the marginalized groups in developing countries such as poor, women, and disabled communities. Almost all countries principally agree with the SMEs' ability to create employment opportunities, specifically for poor, less educated categories, women, and disabled communities, and thereby balance the development with growth and equity (Ifekwem and Adedamola 2016). This is mainly due to the labor-intensive nature of the sector with low entry barriers.

Ifekwem and Adedamola (2016) have further elucidated that SMEs help to maintain the equitable distribution of a nation's income by reducing the gap between the rich and the poor. Moreover, SMEs support mobilizing local resources, conservation and generation of foreign exchange, and even distribution of industries; it also helps mitigate rural-urban migration. The entrepreneurial role of women in SMEs is also an important point that supports SMEs' contribution to achieving balance development with growth and equity. According to IFC (2011), 31% to 38% of SMEs are female entrepreneurs in developing countries. As emphasized by Berezhnytska (2019), SMEs contribute to the economic stability of the country by generating a significant amount of income. Laurenţiu (2016) has further stated that SMEs have an expansive ability that helps evolutions of the middle-class society of countries, enhancing the spirit of private initiative and permitting economic and social prosperity. Therefore, it should be noted that this ability of SMEs has a greater potential to establish a strong middle-class society.

SMEs are remarkably advantageous because they make use of local resources such as raw materials, provide self-employment options, mobilize local savings, and train semi-skilled employees through apprenticeships (Asare et al. 2015). Another distinguishing feature of SMEs is that they are the most efficient source of generating employment (Jeppesen 2005; Singh et al. 2009; Zeng et al. 2010; Nikabadi and Jafarian 2012; Asare et al. 2015; Auzzir et al. 2018). As revealed by IFC (2011), 72%, 20%, and 8% of SMEs perform in the trade and service sector, manufacturing process, and agriculture and other industries, respectively. Furthermore, SMEs are primarily found in labor-intensive industries with low fixed costs of production and minimal entry barriers, and they are believed to primarily create price-taker goods. According to the IFC (2011), women entrepreneurship in SMEs is most prevalent in developing nations, with women owning 31% to 38% of SME businesses in these countries. As stated by many researchers, SMEs have lower job security, quality, and stability than large companies. Furthermore, decreased productivity in SMEs

is commonly noted in the literature, due to an inability to acquire scale merits in industrial operations and several other factors.

*1.2. Defining the Key Terms of the Study—SMEs, SMEs Survivability in the Market, Competitive Strategic Position*

No commonly accepted definition is available for SMEs worldwide. Existing definitions have considered the sector-specific and country-specific characteristics in terms of development status, geographical factors, etc. The main parameters applied in defining the SMEs are the number of employees, annual turnover, capital assets, input usage, production capacity, level of technology adopted, and management practices (Prasanna et al. 2019). The study carried out by the IFC in 2011 among 132 countries globally has revealed that most countries have defined lower and upper thresholds of the definitions of SMEs ranging between 10 to 50 and 50 to 250 workers, respectively. The recent enterprise survey of the World Bank has used the employment size of the firm as 5–19 and 20 to 99 as small and medium size of the enterprises, respectively (World Bank 2017).

In this study, the term "SMEs survivability in the market" is defined as the ability of small and medium firms to continue with the change of mode of economies/business world in the economic competition. The SME adjustment must gain the benefits from the economic competition by facing the three competitive challenges—technology challenges, global challenges, and sustainability challenges—and thereby improve the survivability chance in the competition.

The key term—competitive strategic position—is defined aligned with Porter (1980)'s view on generic strategies. It specifies as a strategy of business firms that could support them to lead firms to a profitable and sustainable competitive business position, thereby enabling them to address the competitive challenges.

## 2. The Theoretical Background of the Study

Porter's generic strategies are applied in this study to determine an organization's strategic positions to acquire a competitive advantage. Michael Porter (1980) stated that, there are four generic strategies based on the market where they compete (narrow or broad) and the source of competitive advantage (differentiation and cost). These strategies are, namely, cost leadership, differentiation, cost focus, and differentiation focus (see Table 1). In some cases, the third strategy is referred to as the focus strategy. This is the organization's choice of competitive scope, and the scope differentiates between the organization's target narrow industry segment or broad industry segment.

Large-scale corporations, as well as SMEs, should assess their strategic position. These companies use Porter's generic strategies as a strategic tool to acquire a competitive advantage. Thus, it is vital to study the strategies these organizations adopt. As stated by Mumbua (2013), differentiation and cost leadership strategies are the best generic strategies for SMEs than differentiation focus and cost focus. Similarly, Dess and Davis (1984) emphasized that cost leadership strategy gives a higher return on assets than differentiation strategy.

Pelham (2000) claims that differentiation strategy has a wider impact on profit than cost leadership strategy. Nevertheless, cost leadership is the most effective approach for SMEs, since the expected objectives of SMEs are increasing profits by reducing setting prices and costs in accordance with the industry average (Julita and Tanjung 2017). As a result of their decreased expenses, these businesses can increase their market share by applying lower prices while still making a significant profit on each product or service they sold.

**Table 1.** Theoretical approach—Porter's generic strategies.

| Generic Strategy | Adopted Strategies by the SMEs |
|---|---|
| Cost leadership | • Develop a Killer Procurement Strategy, Specialize, Boost Efficiency (Gray 2019)<br>• Exploiting the scale of production, experience, standardized products, and efficiency in production (Allmén Sjöberg and Nordström 2019)<br>• Modernizing production and/or implementing process innovations (Leitner and Güldenberg 2010) |
| Differentiation | • Provision of high-quality products (Mumbua 2013)<br>• Offering similar products to their competitors and do not engage largely in consumer marketing (Mumbua 2013)<br>• Focusing on reputation, superior service, support to market their services/products (Mumbua 2013)<br>• Research, development, and innovation (Julita and Tanjung 2017)<br>• Strategic location, reputation, linkage in internal functions, product customization, competitive product mix, superior customer service/support, and flexible pricing (Mumbua 2013) |
| Cost Focus | • Exploit differences in cost behavior in selected segments (Julita and Tanjung 2017) |
| Differentiation focus | • Exploits the special needs of buyers in selected segments (Julita and Tanjung 2017) |

## 3. Review of Literature over Survival Challenges and Survival Strategies Adopted by the SMEs in the Competition

### 3.1. Survival Challenges in the Competition

SMEs heavily contribute to a country's economy. Nevertheless, since they are highly dependent on markets (Butler and Sullivan 2005; Hill and Narjoko 2007), consumers, and suppliers (Nugent and Yhee 2002), continuing their activities in the current environment of economic competition may cause complications. As a result, SMEs experience many challenges when dealing with globalized business activities. Even in favorable situations, SMEs face challenges such as maintaining the existing customers, making long-range plans, and maintaining the payroll (Myles 2010). Manalastas (2009), Hill and McGowan (1999), Yin (1994), and Muranda (2003) have identified various challenges faced by SMEs in developing and developed countries, e.g., barriers from global sourcing, lack of managerial capabilities, lack of financing, recession, low productivity, technology, and regulating burden. SMEs confront significant challenges in sustainable development, such as lack of access to credit, formal business, and social networks (Mboko and Smith-Hunter 2009).

Ifekwem and Adedamola (2016) stated that SMEs face several survival strategies when dealing with businesses. One of the major challenges is the difficulty in attracting funds for SMEs when expanding their businesses. According to them, SMEs are also challenged by issues in power supply, competition/poor patronage, high cost of machine maintenance, inconsistent government policies, high rates of taxes/multiple taxation methods, and other challenges such as infrastructure issues, network issues, and ineffective telephone systems (Ifekwem and Adedamola 2016). Yoshino and Taghizadeh-Hesary (2016) listed the challenges for SMEs in developing countries as having limited access to finance, insufficient databases, low research and development expenditures, not having developed sales channels, and low level of financial inclusion, which they identified as some reasons for the slow growth of SMEs. Alrabeei and Kasi (2014) identified several challenges SMEs are facing at present: constrained access to money and capital market, high rate of small business mortality, shortage of skilled manpower, lack of industrial facilities, poor implementation of policies, poor management practices and low entrepreneurial skills, restricted access to the market place, corruptions, cultural problems, and non-application of research findings and etc.

Similar to the global level, Vijayakumar (2013) stated that there are many SMEs' challenges and failures to occur in Sri Lanka. The inadequacy of organized major industry, shortage of infrastructure and raw materials, civil strife, and governance issues are the constraints to the growth of Sri Lankan SMEs identified by Dasanayaka (2011). The majority of Sri Lankan SMEs are unproductive to grow their business due to delays in decision-making, understaffing, inadequate organization, and poor leading and controlling (Priyanath 2006).

The accordance of the absence of adequate incentives, sexual harassment, the challenge in offering collateral, less chance for low credit facilities, less access to advanced technology, labour scarcity, and involvement in traditional female-type businesses, and acceptance of an "androgynous" style of leadership, are roadblocks faced by female SMEs in Sri Lanka (Wijeyeratnam and Perera 2013). Thrikawala (2011) revealed that SMEs form tight bonds with their families and friends while identifying business prospects. They use social networks to generate support, influence, and initial finance during the start-up period. However, there was a poor relationship with the supportive and sub-firms in Sri Lanka.

Based on several articles, Naradda Naradda Gamage et al. (2020) identified, in a nutshell, seven challenges SMEs deal with globalization. Accordingly, the main challenges are market competition, information and communication technology, global financial and economic crisis, changes in consumer preferences and profiles, global terrorism, multi-national corporations (MNCs), trance national companies (TNCs), and international dumping. Such information proves that although their contribution is vital for an economy, SMEs confront many challenges to survive in the market while dealing with business activities.

### 3.2. Adopted Strategies (Importance of Determining Strategic Positions)

Since SMEs play a vital role in a developing country's economy (Analoui and Karami 2003; Soleh 2008), it is imperative to identify the strategic position of the organization to face the challenges in the marketplace and gain competitive advantage. Currently, SMEs can generate many products in a short time and more effectively. However, SMEs are more vulnerable, and hence, the failure rate of these organizations is very high (Rumanti and Syauta 2013). The Business Statistics Office (UK) identified that 60 percent of SMEs fail in their very first three years after commencement Chak (1998) and Rumanti and Syauta (2013) suggested a proper selection of strategies will undeniably help any type of SMEs to survive in the marketplace. This confirms that SMEs should determine the strategic position of the organization to sustain in the marketplace and face global competition.

According to Borocki et al. (2019), a proper assessment of a firm's strategic position is vital since SMEs face external and internal challenges such as finance and people. Johnson et al. (2008) declared that the strategic position of an organization is determined by the impact of strategies of the external environment, internal resources, and competencies, and based on the expectations and the influence of the stakeholders with interest on the organization. Furthermore, the environmental factors, the strategic capability of the organization, and the purposes bounded with the cultural and political framework of the organization must be considered to understand its strategic position. Adopting market survival strategies is essential for maintaining the sustainability of SMEs in the global economic competition. Ifekwem and Adedamola (2016) suggested that committed and motivated employees, providing orientation and educational programs to change the mindset of business owners, government involvement in the growth, development, and sustainability of SMEs within the country are essential criteria for an SME to survive in the economy. SMEs can follow these factors as survival strategies in the marketplace. Nyanga et al. (2013) stated that diversification of services, maintaining all loyal customers, and increasing the entrepreneurs' psychological strength are significant survival strategies in SMEs in developing nations. These are the most suitable strategies for SMEs to survive in the market while gaining sustainability and facing global economic competition.

For the SMEs' survival, Kitching et al. (2009) identified three broad categories of strategies, i.e., retrenchment strategies, investment strategies, and ambidextrous strategies, while Bamiatzi and Kirchmaier (2014) stated that a focus on cost, differentiation, customization, and internationalization is essential for an SME to survive in the economy and confront competition. As stated by Ndeisieh (2018), technical experience, providing education, and skills to employees, enhancing entrepreneurial and business management skills, enhancing human resource management skills, giving access to external support, bookkeeping and working capital management, research and development, owners' dedication and commitment, marketing strategies, resource-based strategic planning, and owners' understanding

of the business ecosystem, and taxation and regulations will help businesses to survive in the market. According to Iorun (2014), creativity, high risk-taking, and identifying business opportunities are the primary strategies that SMEs should focus on as survival strategies to compete in the global economic competition. He recommended that the country's government should broaden the financial base of these organizations. Interaction of the owners/managers of SMEs should also be promoted via workshops. A government is responsible for providing infrastructure to encourage and promote industrialization, and SME owners should maintain adequate and proper financial records to make appropriate decisions, and focus on research, training, and development programs, develop good personnel management policies, have planned business ownership, and create a stable macro environment, as strategies for SMEs to be sustainable and face competition. He suggested that SMEs operators should focus on effective marketing strategies, such as customer-oriented product lines or services, engaging in creative personal selling, skillful advertising, and having good business location, since they help to smoothen and make the business operations more profitable to survive in the marketplace. Therefore, SMEs can survive in the marketplace by adopting various market survival strategies. This research aims to identify the appropriate marketing survival strategies for SMEs' sustainability in the global economic competition by considering the agro-based industrial entrepreneurs in Sri Lanka.

## 4. Methodology

This study focused on the SMEs in Sri Lanka to identify the factors determining the strategic competitive positions of the SME sector in the developing Asian nations. Like other developing nations in Asia, SMEs make up a large part of the Sri Lankan economy accounting for 80% of all businesses (Rathnasiri 2014). These are found in all sectors of the economy—primary, secondary, and tertiary—and employ persons of different skills, skilled, semi-skilled, and unskilled. Although SMEs play a vital role in the economy, these organizations confront many challenges. The competitive challenges faced by SMEs in the current globalized economy are categorizedas sustainability challenges, technological challenges, and global challenges (Noe et al. 2017). Kankanamge (2014) identified six major challenges associated with Sri Lankan SMEs, namely, competition issues, technology management issues, financial issues, human-resource-related issues, business rules and regulations-related issues, and infrastructure issues. Investigations in the Asian developing nation context also revealed similar challenges (Bisht and Singh 2020). Thus, the strategic position of the SMEs in Sri Lanka would be a typical case for the developing Asian nations in the context of a globalized economy.

### 4.1. Research Design

4.1.1. Study Focus—Agro-Based SMEs

The study narrowed down the research scope in the analysis to agro-based SMEs in Sri Lanka to minimize the sector-wise complexities of the SMEs. The reason for selecting the agro-based SMEs is that research that contributes to the development of an agro-based industrial system may significantly address the issues in the rural economy, such as unemployment, poverty, and low income, because the rural economy is primarily characterized by agriculture. Like other developing nations in Asia, a large portion of the country's population—approximately 70%—lives in the rural sector (DCS 2012). Thus, narrowing down the study scope to agro-based industries would help improve the developing nations in Asia.

4.1.2. Sampling and Data Collection in the Field

As the primary objective of this study is to determine the competitive strategic positions of SMEs in Sri Lanka, the study focused on the deductive method as a research approach. Thus, the study starts with a theoretical preposition outlining the logical connection among concepts and thereby applied a quantitative research technique. The

quantitative data for the study was generated through a field survey administering a pre-tested semi-structured questionnaire which was developed for the major research project funded by the World Bank. The field survey was conducted by the research team under the World Bank-funded AHEAD project—*Assessing the predisposition of small and medium enterprise sector in developing Asian nations towards facing competitive challenges: Case of Sri Lanka*—of Ministry of Higher Education in Sri Lanka, during February to August 2020.

This survey adopted several steps in sampling and data collection in the field. First, the main study was narrowed down to focus on agro-based industries. Here, three agro-ecological zones of the country—Wet zone, Intermediate zone, and Dry zone—formed the base of the sample for data collection since materials for the agro-based industries are primarily supplied from agriculture. Specifically, agro-ecological zones are the main demarcate of the different agriculture productions of the country and the developing nations at large. Second, 30 out of 332 DS divisions were selected by applying systematic random sampling technique. Third, the SME identification survey was conducted in the selected 30 DS divisions to identify the existing agro-based SMEs in both formal and informal settings from November 2019 to February 2020, adhering to the World Bank definition of SMEs used in the Enterprise Survey in 2016. The SME identification survey was conducted primarily to accommodate the non-registered SMEs in the selected DS divisions. Fourth, the study excluded the identified SMEs with less than 1 year experience in their business field to focus on the SMEs with adequate business experience. Fifth, by applying a systematic random sampling method, the study selected 463 agro-based SMEs. The owner/manager of each firm was interviewed by administering a pre-tested semi-structured questionnaire and lasted for 60 to 90 min.

### 4.1.3. Analytical Techniques

By nature, the research approach of the study is deductive as it aims to test the hypothesis related to Porter Generic Strategies in terms of agro-based SMEs in Sri Lanka. Following the theoretical base of the study outlined in the 3rd section of the paper, we identified both strategic and performance variables and employed them in this study to explore the factors determining the strategic competitive position of the agro-based SMEs in Sri Lanka. For this, the study identified the firm's growth in terms of production/business scale as a key performance variable of the sector, which indicates the firm's competitive strategic behavior. The variable could be viewed in the association of other main business firms' performance variables such as profitability, employment growth, and turnover growth. As outlined in Section 2, generic strategic variables were categorized into four sectors: cost leadership, differentiation, cost focus, and differentiation focus.

The study employed a binary logistic regression model to determine the strategic factors which affect the agro-based SME performance and thereby determine the strategic position of the sector in the competition. Table 2 presents the operationalization of the variables incorporated in the logistic regression. The binary logistic regression was used for numerous studies, particularly to determine strategic factors that affect business firms' performance. The model-dependent variable was the firm performance—expanded the business scale ($Y = 1$) and no change or contracted the business scale ($Y = 0$). The logistic model predicts the logit of the response variable (expanded business scale) from the independent variables. According to Gujarati (2003), the likelihood of the SME is in expanding the business scale in the competition, as predicted by odds ($Y = 1$), that is, the ratio of the probability that $Y = 1$ to the probability that $Y = 0$. Then, $P/(1-P)$ is simply the odds ratio—the ratio of the probability that expanded business scale to the probability that no changed or declined business scale during the last five years.

$$Odd\ Y = \frac{p(Y = 1)}{(1 - p(Y = 1))} \tag{1}$$

**Table 2.** Operationalization of variables incorporated in the empirical model.

| Type of Variable | Variable(s) | | Operationalization |
|---|---|---|---|
| Performance variable | Growth of business (production scale) | | Binary variable (1 = increased the business scale during last five years, 0 = no change or decreased) |
| Strategic variables | **Variables under Cost Leadership** | | |
| | (1) | Specialization in production | Dummy variable (1 = adopted the measures to specialize in production during the last five years, 0 = otherwise) |
| | (2) | Experience in the business field | Number of years |
| | (3) | Implementing a process of innovation in product and marketing | Dummy variable (1 = adopted the innovation practices during the last five years, 0 = otherwise) |
| | (4) | Experience in attending business exhibitions in a local setting | Dummy variable (1 = attended business exhibitions at the local setting during the last five years, 0 = otherwise) |
| | (5) | Experience in attending business exhibitions at overseas setting | Dummy variable (1 = attended business exhibitions at the overseas setting during the last five years, 0 = otherwise) |
| | (6) | Gender | Dummy variable (1 = Male, 0 = otherwise) |
| | (7) | Credit market accessibility | Dummy variable (1 = accessed the credit market, 0 = otherwise) |
| | **Variables under Differentiation** | | |
| | (1) | Provision of high-quality products | Dummy variable (1 = Adopt measures to improve the quality of products during the last five years, 0 = otherwise) |
| | (2) | Research and development (R&D) | Dummy variable (1 = Introduced new products to the business during the last five years, 0 = otherwise) |
| | (3) | Strategic firm location | Dummy variable (1 = Urban, 0 = otherwise) |
| | (4) | Strategic market location | Dummy variable (1 = export market, 0 = otherwise) |
| | **Variable under Cost Focus** | | |
| | (1) | Adoption of sustainable business practices | Dummy variable (1 = adopted the sustainable business practices during last five years, 0 = otherwise) |
| | (2) | Adoption of techniques for market efficiency | Dummy variable (1 = adopted the measures to improve the market efficiency of the business during last five years, 0 = otherwise) |
| | **Variable under Focus differentiation** | | |
| | (1) | Focusing on reputation, superior service, and support to market their services/products | Dummy variable (1 = focused on reputation, superior service, support to market their services/products during the last five years, 0 = otherwise) |
| Control variables | (1) | Firm size | Number of workers |
| | (2) | Firm age | Years |
| | (3) | Geographical location | Dummy variable (1 = Wet and intermediate zones, 0 = otherwise) |

The model of this study can be specified as Equation (2). The *logit (Y)* is given by the natural log of odds,

$$ln\left(\frac{p(Y_i = 1)}{(1 - p(Y_i = 1))}\right) = log\ Odds = logit\ (Y) \qquad (2)$$

Equation (2) can be expanded as follows:
With the independent variables, Equation (3) can be expressed as follows:

$$L_i = ln\left(\frac{p}{1 - p}\right) = \beta_i X_i + \varepsilon_i \qquad (3)$$

where $L_i$ is the log odd ratio, $X_i$ is the vector of independent variables (strategic and control variables), $\beta_i$ vector of parameters to be estimated, and $\varepsilon_i$ Stochastic error term.

The log-likelihood ratio help assess the overall significance of the model. Multicollinearity among the independent variables was checked using the Variance Inflation Factors (VIF). STATA 16/IC software was used for data analysis.

## 5. Results and Discussion

### 5.1. Profile of the Surveyed Agro-Based Industries

The descriptive profile of surveyed industries revealed that 72.8% of the surveyed entrepreneurs are males. Most of them are Sinhalese (68.5%), and 27% are Tamils. de de Mel et al. (2009) revealed that ethnicity impacts entrepreneurial activities in Sri Lanka. Most Muslims are engaged in businesses, but they are traders rather than entrepreneurs. As most Sinhalese are Buddhists, the religion of 68.5% of entrepreneurs is Buddhism. The average age of the entrepreneurs is 48 years, and the average age of the surveyed SMEs is 15 years. Most SMEs (73.9%) are located in the dry zone, 20.9% in the intermediate zone, and 23.7% in the wet zone, whereas 66.7% of the surveyed SMEs produce agricultural-based goods and services, and 33.3% make primary agricultural products. In this study, animal farming is also considered a primary agricultural product. A majority (67.2%) of entrepreneurs use modern technological methods for their trades, and 66.1% of the entrepreneurs have registered their businesses. The surveyed sample consists of an average of seven employees.

Most entrepreneurs had GCE (A/L) qualifications, accounting for 42.2%, while 5.9% had postgraduate degrees, and 32.8% had diplomas; very few (0.4%) had no schooling. Thus, most entrepreneurs are educated, which indicates the improved human capital base at the business decision-making level. Most (91.3%) were highly sensitive to the consumers' feedbacks/comments, and only a very few (1.1%) were not sensitive. Regarding participation in the entrepreneurial exhibitions, 40.4% have never partaken in exhibitions, while 56.3% have acquired the technical knowledge related to their businesses through the government, private/international organizations, or non-government organizations (NGOs). Furthermore, 41.5% have never received a loan. Very few SMEs (3.1%) engage with the export activities, and only 16.1% of the total sample have membership in the entrepreneurial associations. Most entrepreneurs (88.5%) are willing to expand their business in the future, and 59.8% face severe competition in the market. This signifies the market potentials for expanding businesses. Most of the SMEs (57.6%) are operating their businesses in the domestic setting, and very few (3%) conduct their businesses at the international level. Table 3 presents the results of binary logistic regression.

Table 3. Results of binary logistic regression.

| Type of Variable | | Variable(s) | Coef. | Std. Err. | Z | *p*>|*Z*| |
|---|---|---|---|---|---|---|
| Strategic variables | | **Variables under Cost Leadership** | | | | |
| | (1) | Specialization in production | −0.0086 | 0.0705 | −0.12 | 0.902 |
| | (2) | Experience in the business field | −0.0449 | 0.0369 | −1.22 | 0.224 |
| | (3) | Implementing a process of innovation in product and marketing | 0.0853 | 0.0496 | 1.72 | 0.086 * |
| | (4) | Experience in attending business exhibitions in the local setting | 0.2123 | 0.0491 | 4.32 | 0.000 *** |
| | (5) | Experience in attending business exhibitions at overseas setting | −0.1190 | 0.1245 | −0.96 | 0.340 |
| | (6) | Gender | 0.1372 | 0.0500 | 2.74 | 0.006 *** |
| | (7) | Credit market accessibility | −0.0058 | 0.0449 | −0.13 | 0.896 |
| | | **Variables under Differentiation** | | | | |
| | (1) | Provision of high-quality products | 0.0431 | 0.0731 | 0.59 | 0.555 |
| | (2) | Research and development (R&D) | −0.0335 | 0.0535 | −0.63 | 0.531 |
| | (3) | Strategic firm location | −0.0328 | 0.0520 | −0.63 | 0.528 |
| | (4) | Strategic market location | −0.2181 | 0.0606 | −3.59 | 0.000 *** |
| | | **Variables under Cost Focus** | | | | |
| | (1) | Adoption of sustainable business practices | 0.1305 | 0.0620 | 2.10 | 0.036 ** |
| | (2) | Adoption of techniques for market efficiency | 0.1134 | 0.0483 | 2.35 | 0.019 ** |
| | | **Variable under Focus differentiation** | | | | |
| | (1) | Focusing on reputation, superior service, and support to market their services/products | −0.0809 | 0.0485 | −1.67 | 0.097 * |
| Control variables | (1) | Firm size | 0.0416 | 0.0296 | 1.40 | 0.161 |
| | (2) | Firm age | −0.0148 | 0.0302 | −0.49 | 0.624 |
| | (3) | Geographical location | −0.0909 | 0.0639 | −1.42 | 0.156 |

Note: ***, **, and * are 99%, 95%, and 90% significant levels.

### 5.2. Variables That Determine the Strategic Position of the Firms

5.2.1. Cost Leadership

- Specialization in Production

Specialization in production means reducing the product diversity and increasing the production of a specific product, which is accompanied by continuing an unchanged production level of the remaining products (Czyżewski and Smędzik-Ambroży 2015). When a firm practices specialization, it will lead the firm to establish an efficient supply chain, greater production possibilities, and achieve economies of scale. Therefore, the firm will be able to decrease the price levels of its products or services because it costs less to produce its goods and to facilitate its services. This ultimately helps the firm raise as a cost leader by selling more units at a lower profit margin and gaining a competitive advantage in the globalized market. In the present study, however, specialization in production is recognized as an insignificant variable under the cost leadership strategy (*p* = 0.902).

In the Sri Lankan agribusiness sector, several constraints make this variable insignificant. Usually, lack of business resources hinders the implementation of all steps related to engaging a new strategic change simultaneously, like the specialization of production, especially in SMEs (Mendes and Lourenço 2014; Saleh and Ndubisi 2006; Shields and Shelleman 2015). Hence, SME owners must evolve a solid and integrated implementing plan

consistent with firms' elements for executing a specialization strategy. When focusing on a narrow range of products through specialization, SMEs must predict the ability to perform in the selected arena. Moving forward without proper planning may result in disappointed consumers, generating a credibility gap between the firm and the chosen arena.

Moreover, the product specialization process must recognize the available resources, abilities, and skills required (Borland and Yang 1992; Nadube and Didia 2018). When an SME owner practices product specialization, it is necessary to promote it to multiple target markets, and specific types of consumers may prefer specific product features and benefits. SMEs in developed countries invest a significant amount of money and resources for implementing ad campaigns in markets to gain a competitive advantage in penetrating diverse markets. As a developing nation, it is difficult for Sri Lankan SMEs with poor financial bases to implement attractive marketing campaigns. These could be possible reasons for identifying such factors as insignificant among the Sri Lankan agro-based SMEs.

- Experience in the Business Field

Cost leadership is considered a strategy to make the firm a cost leader among other firms by reducing overhead expenses, establishing effective production facilities, and eliminating marginally profitable customers. Having experience in business means the firm knows its range intimately. In developed countries, experienced SME owners know the changing market demands and trends in their industries. However, applying the dimension of experience in the business field is identified as an insignificant factor for the agro-based SME organizations in Sri Lanka ($p = 0.224$).

According to Chang (1997), if a firm manages to learn from the advantages from experience curve well, it may lead to price competitiveness by reducing overhead expenses and average unit cost that easily attract their target audiences. These advantages occur from the cumulative knowledge of a firm over time in learning to enhance its business operations. Entering markets become more complex and costly for a new firm because already established firms in the market have attained a good experience, higher market share, and low per-unit production costs. Compared with earlier entrants in the market, a new entrant with low experience has considerably delicate cost positions due to its small market share.

In the SME sector of Sri Lanka, Priyanath and Premaratne (2015) revealed that 70% of businesses failed within three years of commencement and, among them, 60% failed within the first year of commencement. This is mostly due to the fewer experience in decision-making and less potential of businesses to face the competitive challenges (Gilmore and Carson 2000). In the cost leadership strategy, cost cuttings never occur easily or automatically—firms must work hard for them with their experiences. Hence, the experience in the business field is vital for Sri Lankan SMEs who are following cost leadership strategies to take appropriate decisions at the right time to remain competitive in the globalized market. If there are supportive hands at the commencement, most firms have greater potential to attain many improvements from their business experiences over time.

- Implementing Process of Innovation in Product and Marketing

The study results provide empirical evidence that the implementing process of innovation in product and marketing ($p = 0.086$) performs a positive and significant effect on the cost leadership strategy. The innovation process is a key factor that marketing strategists use to win markets and customers through the development of competitive advantage, and it guides to a process of modification in an organization and its market offerings (Kanagal 2015). Conventionally, the systematic process of innovation has been in the authority of the research and development institutions. However, with the emergence of the internet and the world wide web, the 'open innovation' dominated, where seekers and problem solvers come together; consumers could also participate.

Innovation is the introduction of new methods or combinations, instead of older ways of converting inputs into the output to fabricate changes in the comparison between use price and value provided to obtain customer satisfaction (Brines et al. 2013; Okpara 2007).

Although cost leadership is highly effective in acquiring high market share and bringing out customer attraction, it is difficult to deploy. Hence, a better and well-planned implementing innovation process in product and marketing is a major deal in cutting costs. In short, the better the innovation used by a business, the more are its possibilities of dwelling as a cost leader in the long run.

Sri Lanka remains behind similar-income countries; hence the country should focus more on introducing new technologies, enhancing the business environment, increasing the technology transfer, and improving firms' strategy and operational efficiency for a better implementation process of innovation in product and marketing. External assistance is vital for a successful implementing process, as Sri Lanka's agro-business sector generally suffers from a lack of marketing knowledge, technical skills, and training resources (Asian Development Bank 2006; Stamm et al. 2006). Hence, vendors and external consultants are the key sources of external innovative knowledge and skills in firms. Thus, higher-level vendors, external consultants, and support will increase the effectiveness in the implementing process of innovation in product and marketing. However, the attitude and orientation of the owner of the firm is also a key to implementing innovativeness within SMEs. Therefore, SMEs that pursue a cost leadership strategy could carry out a better organizational performance with a good implementing process of innovation.

- Experience in Attending Business Exhibitions in the Local Setting

Among the significant variables identified under the cost leadership strategy, one major variable is the experience of attending business exhibitions in a local setting ($p = 0.000$). The exhibitions create competitive platforms for SMEs to sell their products and find new business avenues. SMEs can tirelessly inspect and learn the ways and methods of sustainably cutting costs below those of other competitors by evaluating them through these exhibitions. Hence, a firm's interactions with its business environment are essential for its survival in the competition.

In the Sri Lankan agro-business context, local exhibitions create a powerful marketing medium and bring together many buyers and sellers in a single stage in a limited space of time. Exhibition series such as "Enterprise Sri Lanka" and "Deyata Kirula" organized by the Ministry of Finance, Ministry of Industries, and other authorities in the last decade created platforms for intended SMEs to share knowledge and obtain the know-how on cost leadership strategies. These opportunities did not merely gather all the significant characters from the industry but allowed them to inspect and evaluate various products, compare, and acquire vast knowledge on various strategies used. Hence, it is inevitable for the SMEs following the cost leadership strategies to obtain brand recognition in the Sri Lankan market by attending local exhibitions. Similarly, these local exhibitions provide customers with a comprehensive overview of the entire market and industry. If businesses that follow cost leadership strategies can promote standard products at an affordable price on these platforms, then it becomes one of the most preferred products among local consumers.

- Experience in Attending Business Exhibitions at Overseas Setting

Attending business exhibitions enables local SMEs to acquire knowledge on specific cost leadership strategies, strengthen business relationships, and find new opportunities in the globalized market. It allows local SMEs to experience and inspect technologies needed to reduce their production costs and increase production efficiency on their own. Hence this is identified as an important variable in cost leadership strategy in developed countries. Nevertheless, the present study revealed that the Sri Lankan agro-business sector recognizes this as an insignificant variable ($p = 0.340$).

According to the previous variable, even though attending business exhibitions in local settings is a common culture among Sri Lankan agro-based SMEs, gaining experience in attending business experience in overseas settings became very rare and difficult with several barriers. With the poor financial base, almost all SMEs are unwilling to participate in overseas exhibitions. Hence it is the responsibility of the institutional setting and the

government authorities. Most international level trade shows and exhibitions arrange seminars, lectures, practical sessions series, and presentations on newer strategies with the help of leading experts in the field. This study proves that Sri Lanka is still not ready to draw the benefits from these events. Even though some institutes like the District Chamber of Commerce are involved in organizing foreign tours to visit exhibitions, the study results revealed that such measures are insufficient to cause a significant effect on the subject.

- Gender

Currently, many firms are aiming to become cost leaders, seeking profits from the competition. These firms are vigorously following low-cost strategic management practices to minimize costs and penetrate new market areas. The study revealed that gender plays a prominent role ($p = 0.006$) in sustaining and gaining inherent advantages over competitive firms to ensure profitability and long-term success as cost leaders. These observations proved that gender clearly impacts the cost leadership strategy of SME firms.

The factor "gender" has a unique nature within firms because of its specific traits. According to Quan (2012), females have less entrepreneurial intention than males, measured as impulsive intention or deliberate intention. Moreover, his study revealed that females are less optimistic and less likely to take risks than males in the business environment.

In most parts of Sri Lanka, men are associated with many aspects of agro-based industries. They are more likely to take risks, expand businesses, perform, and are more optimistic than females in following cost leadership strategies. Some female-led businesses are managed differently than males, mainly because of differences in their abilities, attitudes, education, experience, and socialization. Some of their economizing abilities made them more survivable in the competition.

Females enter business not only for financial gain but also to achieve intrinsic goals such as flexibility to interface with family, independence, and work commitments (Winn 2004). The role of women in entrepreneurship in the country's cultural context has always been disregarded from sociological perspective (Hewapathirana 2011). According to Watkins and Watkins (1984), female entrepreneurs tend to suffer frequently from various problems, including family conflicts, and sometimes by husbands who are actively hostile to wives' businesses, though most male SMEs have supportive wives. According to Nishantha (2009) males are more inclined than females to have an entrepreneurial mindset. Further research discovered that Sri Lankan entrepreneurs have psychological attributes such as the desire for accomplishment, internal locus of control, and risk-taking attitude. Langworthy (2018) identified that gender related issues, social norms, and institutional discriminations are the factors that hamper the ability to start-up and grow businesses for female SMEs in Sri Lanka.

Generally, it is accepted that women have strong financial and attitudinal strategies than men to grow as cost leaders (Lerner and Almor 2002). These different abilities and aspects in gender will ultimately support the firms to be cost leaders in the long run and face dynamic business settings.

- Credit Market Accessibility

Credit accessibility of SMEs is considered a key factor in reducing a firm's costs of capital and maximizing low-cost products. According to Beck et al. (2008), developing a country's financial sector plays a vital role in enhancing SMEs' growth. In the developed countries with high-level credit market development, businesses have relatively easy access to credit funds, thereby achieving low-cost strategies with financial stability. According to the present study, credit market accessibility is identified as a nominal dimension for the agro-based SMEs in Sri Lanka ($p = 0.896$).

In a developing country like Sri Lanka, the quality of the credit market and other institutions is low compared to developed countries (Gamage 2003). This makes it very difficult for SME owners to find favorable processes to obtain loans and reduce costs to achieve cost leadership strategies as they have to follow lengthy documentation procedures, and laws and regulations. The inability of SMEs to find guarantees, long delays, less access

to the bank system, and lack of knowledge of procedures are some incompatibilities that arose for SMEs when accessing the credit market (Gamage 2003). Hence, facilitating finance is considered a risky financing option for lenders in the credit market, and it creates difficulties in obtaining credit facilities for SMEs from formal financial institutions. As a result of the factors described above, most SMEs cannot receive financial support on their own. Hence, they have to approach risky informal financial sources such as microfinance institutes and ultimately fall into debt traps. Thus, though most developed countries consider credit market accessibility as one of the important aspects of the cost leadership strategy, the Sri Lankan SMEs are highly limited with restrictions. Hence, the present study identifies credit market accessibility as an insignificant factor for agro-based SMEs in Sri Lanka.

5.2.2. Differentiation

Product differentiation is considered to be the act of delivering products to the customers in ways different from one another. According to McGee et al. (2010), differentiation is the characteristic of imperfect markets, which prioritize non-price strategies.

- Provision of High-quality Products

The present study disclosed that the provision of high-quality products is a nominal dimension ($p = 0.555$). Quality can be explained as product features with the ability to meet customer requirements and enhance customer satisfaction (Eldin 2011). High-quality products can attract more customers who are profoundly concerned about the product quality (McGee et al. 2010). The quality can be identified as a source of differentiation advantage.

Widuri and Sutanto (2018) noted that using a differentiation strategy requires the right technology, specialized assets, and high-knowledge workers to differentiate their products from the competitors. Evidently, technology plays a vital role in the differentiation strategy. Sri Lanka, being a developing country, lacks the developed technology that can use for the production processes (Pushpakumari and Watanabe 2009). This could be one of the reasons to consider the high quality of products as an insignificant dimension.

According to Gamage (2003), most developing countries need less capital and less infrastructure for developing their SME sector. Even though this has helped the SMEs to be successful to some extent, they do not have sufficient capabilities to produce high-quality products due to the lack of infrastructure facilities compared to other countries. Such observations signify that some characteristics of SMEs can lead to a quality reduction of their products. This discussion elaborates the potential reasons for the non-applicability of the differentiation strategy in the Sri Lankan SMEs.

- Research and Development

Research and development are vital for SMEs to be innovative and enhance survivability in the increasingly competitive environment (Erickson and Jacobson 1992; Ghaffar and Khan 2014). Research and development affect the firm performance, and it improves productivity (Romer 1986) and performs a key role in differentiation strategy when delivering unique products to consumers. However, in the present study, research and development are recognized as insignificant among the Sri Lankan agro-based SMEs ($p = 0.531$).

Mosey (2005) stated that SMEs obtain experience and dependability by diverting their ideas into new aspects of research and development, and hence they can experiment with new product developments to meet emerging specific and unique consumer needs. Many product differentiators in developed countries have the financial stability to invest a considerable amount of their resources in research and development to achieve a competitive advantage over other firms by producing excellent quality products, achieving superior innovations, and prioritizing unique customer needs.

The Sri Lankan agro-business environment has several barriers that hinder the contribution of research and development to achieve the differentiation strategy. The lack of a strong financial base is recognized as a significant constraint to SMEs in developing countries like Sri Lanka (Amaradiwakara and Gunatilake 2017). Similarly, economic and

legislative barriers cause lagging in evolving a suitable environment for the effective functioning of these strategies. Therefore, although research and development are recognized as important factors in differentiation, they are recognized as insignificant due to many country-level restrictions in the Sri Lankan context.

- Strategic Firm Location

One of the main dimensions the researchers identified with regards to the differentiation strategy is the strategic firm location. According to Piga and Poyago-Theotoky (2005), differentiation strategy is more connected with the firm's location as it can support consumers to have a direct and quick connection with the business. Hence, customers are willing to pay a higher price for the convenience they experience. Even though this is considered as one of the significant factors in the global context, applying the dimension of strategic firm location is considered an insignificant factor for the agro-based SME sector in Sri Lanka ($p = 0.528$).

Thus, it is vital to identify the factors that impact the strategic firm location to be insignificant in agro-based SMEs organizations in Sri Lanka. The classical location theory suggests that the firm's location can lead the business organizations to maximize profit. According to Dixit et al. (2019), strategic business location decisions can depend on economic factors such as transportation costs, tax incentives, proximity to the suppliers and customers, and the sustainability aspects.

Due to several legislation barriers that the Sri Lankan SMEs face, the strategic firm location could be identified as an insignificant factor in achieving the differentiation as a business strategy. Gamage (2003) explained that the SMEs industry failures in Sri Lanka are mainly due to the government's economic policies. The locations of firms in the agro-based industries must face several legislation barriers when establishing the firms. The strict policies such as taxes and other economic factors have limited the establishment of businesses where they have accessibility to customers (Rohlin et al. 2014).

Even though the developed countries consider the strategic firm location as one of the important aspects in the differentiation strategy, Sri Lankan SMEs, as in a developing country, are highly limited by restrictions and thus regard this factor as insignificant for the agro-based SMEs.

- Strategic Market Location

The variable, strategic market location, is only statistically significant under a differentiation strategic position. Product differentiation is considered as the act of delivering products to customers in a way differing from one another. The present study identified that the strategic market location ($p = 0.000$) plays a vital role in the differentiation process than the other identified dimensions. The strategic market location could be defined as 'how and where a company will place the products and the services that the organization is manufacturing to lead the consumers to purchase them and increase the market share.' It has a unique role to play in terms of the differentiation strategy. If an organization is moving towards the differentiation strategy where it is necessary to be unique, the strategic market location they hope to address should also be unique compared to the competitors of the company. The firms can select a specific market location where they can attract more customers and sell their products in an exclusive manner to gain a competitive advantage.

The SMEs can focus on the differentiation strategy to face the competition and survive in the dynamic business environment as they are more vulnerable to challenges. Hence, SMEs should select a strategic market location and deliver the product and services they are offering to that specific market segment. If the organization is focusing on the differentiation strategy, the present study suggests that it will reduce the closed down rate of the SMEs due to several challenges they face over the economic shocks. Even though the agro-based SMEs in Sri Lanka contribute to the economy in a high percentage, these companies face challenges because of this vulnerability. Hence, these firms can focus on the differentiation strategy by selecting the strategic market location where they can attract more customers and deliver unique products for the customers, compared to their competitors in the

business environment. As most agro-based companies are SMEs, which are labor-intensive, firms can consider employees' innovative thinking and study the strategic market locations to perform better than the competitors and survive in the competitive business world.

### 5.2.3. Cost Focus

Cost focus is another critical variable identified in Porter's Generic strategies. According to the researchers, cost focus can be defined as an attempt for firms to concentrate on implementing targets on one or more specific customer segments or niche markets. The firms tend to rely on both the differentiation and the cost leadership strategy. It is not a hybrid of the two strategies, as there should be a specific separation between them. The cost focus is applied chiefly to the narrow market segments where the firms should concentrate more.

- Adoption of Sustainable Business Practices

According to the present study, the adoption of sustainable business practices has become significant ($p = 0.036$) among the other variables. The adoption of business practices is identified as firms focusing on productions and services and delivering them using sustainable business practices.

Sustaining the environment is considered a vital aspect in the present-day scenario (Chang and Slaubaugh 2017) because human behavior has created a negative impact on the world at present. With the adaptation of sustainable business practices of the eco-friendly aspects, the firms can utilize the resources without compromising the needs of future generations. The firms can focus on narrow markets by considering the focus as a strategy so that the firms will perform the business activities in a more specific aspect without harming the environment.

The SMEs in Sri Lankan agro-based industries will gain a competitive advantage over the competitors, leading them to survive in the dynamic business environment. More importantly, as the firms are highly focused on a niche market, they will gain the opportunity to be more conscious about sustainable business practices. As discussed in the study, SMEs are more vulnerable to competitive challenges than other business organizations. If the organizations adopt sustainable business practices, they could overcome the challenges and perform well in facing the competition.

In the Sri Lankan context, most agro-based industry entrepreneurs are moving towards sustainable business practices where they identify the opportunities through the window-and-the-corridor principles. Moreover, the agro-based organizations can adopt sustainable practices such as maximizing material and energy efficiency, adopting proper waste management systems, and recycling and renovating, which can enhance the firm's performance and competitive advantage. This will ultimately support the organizations to be beneficial in the long run and encounter dynamic business settings.

- Adoption of Techniques for Marketing Efficiency

Among the significant variables identified under the focus category, the adoption of techniques for marketing efficiency ($p = 0.019$) is another critical variable. The adoption of techniques for marketing efficiency can be explained as the firms that use technology to make the marketing function more efficient. Like the adaptation of sustainable business practices, this variable also highly focuses on the narrow markets where a clear separation is expected between cost leadership and differentiation. That is, the firms must concentrate on the arrangement of the organizational performance to gain a competitive advantage and survive in the dynamic business environment.

According to Ansari and Mela (2003), the internet plays a significant role in SME marketing consisting of access to new markets, business-to-business collaborations, improving internal efficiency, and many more. As per Kara et al. (2005), the researchers have identified a strong relationship between marketing efficiency and technology adoption. The past studies have revealed that market orientation helps organizations to identify customer

requirements, to share information across the organization, and, more importantly, to have a better competitor analysis for the organization.

SMEs gain the ability to enhance an organization's profitability by adopting technological developments to make the marketing function of the organization more accurate and convenient. The organizations will increase the firm's transparency by evaluating the competitor analysis via the internet and other technological resources. This leads the organizations to share the information conveniently between the individuals in the organizations. Similarly, agro-based industrial entrepreneurs could benefit from adopting technology in improving the efficiency of marketing functions. The entrepreneurs can use several technological developments such as social media networks, e.g., Facebook, Instagram for promotional campaigns, as well as Cloud computing for better customer-supplier relationships in exporting products and importing the raw materials, etc.

Barriers to adopting technology should be identified in the process of making marketing function more efficient in the Sri Lankan context. If the agro-based industrial entrepreneurs focus on using technology development and engage with specific promotional companies, they will easily capture the focused narrow market. This will help them face the competition in a well-developed manner and gain a competitive advantage to survive in the highly volatile business environment.

### 5.2.4. Focus Differentiation

Another strategy the firms have identified is focus differentiation. Focus differentiation can be defined as organizations are targeting the narrow market and differentiation as the strategy. In other terms, it is an approach the firms use to gain a competitive advantage over the competitors by providing a superior product or a service to the customer. In this category, the target market is a small and specified segment. High prices could be a significant factor in this strategy.

- Focusing on Reputation, Superior Service, and Support to Market their Services/ Products

The only variable representing focus differentiation concentrates on reputation, superior service, and support to market their products or the services. The research findings consider this as a significant variable ($p = 0.097$).

When an organization targets a specific market and focuses on the differentiation strategy, they should implement strategies that allow them to focus on the company's reputation and its goodwill, as it addresses a significant market segment. The firms must ensure that they provide superior service to the customers who are willing to pay a higher price since they expect a unique product/service compared to other competitors in the marketplace. Additionally, the firms must deliver the product or services that support marketing their products/services.

The SMEs are more concerned with focus differentiation as they cannot provide products or services to a broader market. In that case, they focus on a specific market segment and attempt to be more conscious about the customer requirements to capture the market by attracting new customers and retaining the existing customers. The SMEs can provide a superior service to the customers and secure the reputation without harming their expectations to challenge the competitors and gain a competitive advantage to survive in the marketplace. Due to the specific features of the SMEs, such as vulnerability and labor-intensive, the firms must aim for an unblemished reputation and adopt necessary strategies to deliver a superior service to the specified market segment.

In the Sri Lankan context, especially in the agro-based industry, entrepreneurs should focus on their reputations. The service they expect to deliver to the customers should be at a superior level that supports marketing their products or the services. They can rely on producing unique products even at a higher price but should maintain the quality standards. This allows them to match the customer requirements in a specific manner. The entrepreneurs can adopt quality tools to improve the standards, learn mechanisms to reduce wastage, and implement necessary strategies to market their products to the

selected customer segments. In that way, the agro-based entrepreneurs will enhance their performance and gain a competitive advantage by facing the competition more successfully.

## 6. Concluding Remarks

Understanding the competitive strategic position of a business is highly essential to firms or specific business sectors to survive in the economic competition and to make necessary adjustments. This study assessed the factors that determine the competitive strategic position of SMEs in the agro-based sector in Sri Lanka. The contribution of the study could be deliberated in two ways: theoretical contribution and managerial implications.

### 6.1. Theoritical Contribution

Competition on any scale of the business world is a principally accepted norm in the globalized economy. As outlined at the outset of the research, theories which are centred on competitive strategic positions of the business firms or specific business sectors, establish a base for the studies in the field. The variables, which applied to determine the competitive strategic positions of the SME sector, were identified based on the theoritical and applied literature. The study's findings revealed seven strategic variables: innovation in product and marketing, business exhibition in the local setting, gender, strategic market location, sustainable business practices, marketing efficiency, and business reputation and superior services. These variables determine the competitive strategic position of agro-based SMEs in Sri Lanka. Even though these factors were found significant in the Sri Lankan agri-business context, the study identified and commented on seven insignificant variables: specialization in production, experience in the business field, experience in attending business exhibitions at overseas setting, credit market accessibility, provision of high-quality products, research and development, and strategic firm location—developed countries recognize these as significant variables.

### 6.2. Managerial Implications

The strategic variables which are statistically significant could be recognized as the variable which improve the firms' survival and competitive conditions in the free-market era. First, implementing a process of innovation in product and marketing, experience in attending business exhibitions at local settings, and gender impact the cost leadership as those variables are specific to attract competitive business advantages via cost leadership. Specifically, the implementing process of innovation in product and marketing must address the changes in consumer preferences. The experience gained by attending business exhibitions in the local setting provides SMEs the opportunities to broaden their understanding of new trends in product markets in terms of technology and product market. It also proved that both men- and women-specific entrepreneurial characteristics lead firms to attain several specific and positive approaches to survive the intensified competition. Specifically, evidence demonstrates that developing Asian nation women entrepreneurs are primarily in risk-averse positions in business competition.

Second, the variable-strategic market location—plays an essential role in offering products to a specific market segment. It also improves the survival chance of the SMEs in the competition as firms could attract more customers and deliver unique products for the customers than the competitors.

Thirdly, adopting sustainable business practices is the new strategic way of attracting customers under cost focus. These practices concern the environment, human health, and resource sustainability in the production and consumption process, and they add production and marketing values for their businesses. Adoption of techniques for marketing efficiency also contributes to functions of marketing to be more efficient. There is much evidence that most SMEs fail within a shorter period of their business commencement due to the failure in marketing.

Fourth, business reputation, superior services, and support to market services and products are recognized as an important strategic variable. It attracts new customers to the

business and retains existing customers, thereby improving the SMEs' competitiveness in the free market condition.

Finally, in case of Sri Lankan agro-based SME sector, seven variables, which are recognized as significant variables in terms of developed countries, were identified as insignificant in determining strategic competitive position of the SMEs. Therefore, the government and policymakers must initiate measures to establish a more favorable business environment for SMEs to gain competitive advantage from these variables in the near future to permit an ameliorated and strong SME sector in Sri Lanka as, like other business sectors, the SME sector must also face the globalized business environment. For strengthening the country's SME sector, organizing customized supportive programs, restructuring of institutional setting, policy adjustments, and reforms are vital to assure present and future goals of the industrial sector. In particular, the accountable, interconnected, private, and government institutional setting in the country must establish answers to these unanswered issues.

The findings of this study could be generalized to other developing Asian nations since agriculture and agro-based industries in these countries have similar characteristics. However, the study highlights a few directions for further studies in the field, specifically to answer the questions: why and how the seven variables were insignificant in the context of Sri Lanka and what policies could be implemented to improve the survival conditions of the SMEs in the economic competition.

**Author Contributions:** Conceptualization, J.J. and R.P.; methodology, S.K.N.G. and P.R.; investigation, P.R.; resources, S.K.N.G.; writing—original draft preparation, E.E., B.S., J.U. and G.A.; writing—review and editing, R.P., J.U. and B.S.; supervision, J.J.; project administration, R.P.; funding acquisition, J.U. All authors have read and agreed to the published version of the manuscript.

**Funding:** This research was supported by the Accelerating Higher Education Expansion and Development (AHEAD) Operation of the Ministry of Higher Education, funded by the World Bank.

**Institutional Review Board Statement:** Not applicable.

**Informed Consent Statement:** Not applicable.

**Data Availability Statement:** Not applicable.

**Conflicts of Interest:** The authors declare no conflict of interest. The funders had no role in the design of the study in the collection, analyses, or interpretation of data, in the writing of the manuscript, or in the decision to publish the results.

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
