# Peer review of "Factors Determining the Competitive Strategic Positions of the SMEs in Asian Developing Nations: Case Study of SMEs in the Agricultural Sector in Sri Lanka"

_economies, doi:10.3390/economies9040193_

Round 1
Reviewer 1 Report
The research area is topical and interesting, but the topic itself needs to be corrected. In my opinion, its second part can be summarized as follows: "..... case study of an SME in the agricultural sector in Sri Lanka". In my opinion, too much attention has been paid to globalization in the introduction. There is no need to define it because it is taken as a fact and a commonly known macroeconomic paradigm. The synthetic and factual characteristics of SMEs in Asian countries are very well presented, especially their impact on the socio-economic environment. It is also worth mentioning their influence on the shaping of the middle class in society, which plays a decisive role in building the prosperity of the state. The consistency of the theoretical assumptions - Porter's strategy with the methodology and results is worth noting. In the literature review section, reference should be made to the reports on SMEs in Sri Lanka. The presentation of publications on Asian developing countries is fully justified, but no details about the analyzed country are available. The methodological assumptions are well-perceived, but there is no broader description of the initial research method. The authors write in section 4.1.2 that they used survey data. A survey, whatever its name, is a tool, not a method or technique. Here it is necessary to describe the quantitative research and precisely name it (the survey method), the research technique (questionnaire research) and a short description of the tool - questions that are a consequence of the variables adopted for the study. I highly appreciate the analysis of the research results and their reference to the economic situation in Sri Lanka. It is worth presenting selected - general information about SMEs in Sri Lanka in gender describing the condition of this type of enterprise in developing countries in Asia.
Author Response
Firstly, I would like to thank the editor and two reviewers for reviewing the manuscript using your valuable time and making valuable and constructive comments. We herewith kindly inform you that we did our best to improve the paper by taking into accounts your valuable comments.
Please see the attachment.
Thank you.
Sincerely yours
Corresponding author

Reviewer 2 Report
The paper addresses an important issue of determining critical antecedents of the competitive strategic position in the Asain developing nations. Relevant analyses methods have been adopted. The paper has several contributions. However, I have several suggestions to further strengthen the paper's arguments.
First, though the introduction section critically introduction the under investigation phenomenon, however, the section lacks to highlight the theoretical basis. I suggest you to mention at least one existing theory that can explain the topic under examination.
Second, I suggest you to segregate the discussion and conclusion into Discussions (i.e., theoretical contribution, managerial implications, and limitations and future directions), and a brief conclusion section. By doing so, the reader will clearly delineate the main contributions to the theory and implications for practice.
I wish you all the best for your research!
Author Response
Firstly, I would like to thank editor and two reviewers for reviewing the manuscript using your valuable time and making valuable and constructive comments. We herewith kindly inform you that we did our best to improve the paper by taking into accounts your valuable comments.
Thank you.
Sincerely yours
Corresponding author
